# The Effect of Continuous Low-Intensity Exposure to Electromagnetic Fields from Radio Base Stations to Cancer Mortality in Brazil

**DOI:** 10.3390/ijerph18031229

**Published:** 2021-01-29

**Authors:** Nádia Cristina Pinheiro Rodrigues, Adilza Condessa Dode, Mônica Kramer de Noronha Andrade, Gisele O’Dwyer, Denise Leite Maia Monteiro, Inês Nascimento Carvalho Reis, Roberto Pinheiro Rodrigues, Vera Cecília Frossard, Valéria Teresa Saraiva Lino

**Affiliations:** 1Sérgio Arouca National School of Public Health, Oswaldo Cruz Foundation, Rio de Janeiro 21041-210, Brazil; monicakra@gmail.com (M.K.d.N.A.); odwyer@ensp.fiocruz.br (G.O.); inesreis@ensp.fiocruz.br (I.N.C.R.); verafrossard@gmail.com (V.C.F.); valeriaslino@gmail.com (V.T.S.L.); 2Institute of Social Medicine, Rio de Janeiro State University, Rio de Janeiro 20550-900, Brazil; 3Instituto Metodista Izabela Hendrix, Belo Horizonte, Minas Gerais 30160-012, Brazil; mreengenharia@mreengenharia.com.br; 4Medical Sciences College, Rio de Janeiro State University, Rio de Janeiro 20550-170, Brazil; denimonteiro2@yahoo.com.br; 5Federal Center for Technological Education Celso Suckow da Fonseca, Rio de Janeiro 20271-110, Brazil; roberto.88rpr@gmail.com; 6Petrobras, Rio de Janeiro 20031-912, Brazil

**Keywords:** cancer, mortality, electromagnetic fields, breast neoplasms, lung neoplasms, esophageal neoplasms, uterine cervical neoplasms

## Abstract

Background: this study aims to estimate the rate of death by cancer as a result of Radio Base Station (RBS) radiofrequency exposure, especially for breast, cervix, lung, and esophagus cancers. Methods: we collected information on the number of deaths by cancer, gender, age group, gross domestic product per capita, death year, and the amount of exposure over a lifetime. We investigated all cancer types and some specific types (breast, cervix, lung, and esophagus cancers). Results: in capitals where RBS radiofrequency exposure was higher than 2000/antennas-year, the average mortality rate was 112/100,000 for all cancers. The adjusted analysis showed that, the higher the exposure to RBS radiofrequency, the higher cancer mortality was. The highest adjusted risk was observed for cervix cancer (rate ratio = 2.18). The spatial analysis showed that the highest RBS radiofrequency exposure was observed in a city in southern Brazil that also showed the highest mortality rate for all types of cancer and specifically for lung and breast cancer. Conclusion: the balance of our results indicates that exposure to radiofrequency electromagnetic fields from RBS increases the rate of death for all types of cancer.

## 1. Background

Mobile phones have become extremely common in modern times. Wireless technology has a large number of Radio Base Stations (RBSs), which transmit information through radiofrequency signals. In 2006, there were already more than 1.4 million RBSs in the world [1]. In the Brazilian capitals, RBSs were implemented in 1992 in Brasília (the capital of Brazil), and in 2017, there were 27,145 RBSs indexed in the capitals [2].

The effect of electromagnetic fields emanating from RBS on health is not very well known. The World Health Organization (WHO) reported, in 2006, that scientific knowledge indicates that RBS radiofrequency exposure is within the international standards and, therefore, does not pose a risk to human health [1]. However, in 2014, the WHO recognized the need to promote research to investigate the effect of the radiofrequency field on human health as a priority in order to fill the knowledge gaps [3]. Several issues relating to new wireless technologies are currently highlighted: the environmental impact of RBS radio frequency exposure, its effects on human health, its thermal effects, and its noise emission [4].

In Brazil, the National Telecommunications Agency (ANATEL) is the entity that regulates the electromagnetic emission of RBSs in accordance with the limits established by Resolution No. 700 of 28 September 2018 (Union Official Diary) [5]. In addition to ANATEL, telecommunication antenna installations are also regulated by municipal laws in order to minimize environmental and human health impacts [4].

Mobile phone-derived electromagnetic fields are classified by the International Agency for Research on Cancer (IARC) as possibly carcinogenic to humans [3,6,7]. The intensity of the RBS radiofrequency fields is higher near the antenna and decreases as the distance from it is greater [1,8]. In big cities, however, RBSs are located very close to populated areas, above or between homes and businesses. The antennas are so close to homes that the multi-story windows of residential buildings, for example, are side by side to these antennas [9].

Despite the scarce knowledge on this subject, there are few resources allocated to investigating the role of exposure to electromagnetic fields from RBSs on human health. In the United States, for example, until 2010, no funding had been reserved by government agencies to study the possible health effects on people living near RBSs [9]. This study aims to estimate the rate of death for cancer according to RBS radiofrequency exposure, especially by breast, cervix, lung, and esophagus cancers, which are among the most common cancers in Brazil for men, women, or both sexes.

## 2. Methods

This is an ecological study using capitals as the unit of analysis. We collected information on the number of deaths by cancer per gender, age group, Gross Domestic Product (GDP) per capita, death year, and the amount of exposure over a lifetime.

Information on deaths by cancer per gender and age was collected from the Mortality National System (SIM) from the Computer Science Department of the Unified Health System (DATASUS) website [10]. We investigated all cancer types and some specific types: (1) deaths by breast cancer (International Classification of Diseases 10th Revision (ICD10) group—malignant breast neoplasms), (2) deaths by cervix cancer (C54 category of ICD10—malignant neoplasm of the cervix), (3) deaths by lung cancer (C34 category of ICD10—malignant neoplasms of the bronchi and lungs), and (4) deaths by esophageal cancer (C15 category of ICD10—malignant neoplasm of the esophagus). The choice of these specific types of cancer for this study took into account the high frequency of new cases in women or in men. Current statistics from 2020 from the National Cancer Institute show that breast cancer had the highest number of new cases in 2020 for women (about 66,000 cases, corresponding to about 30% of cases). Cervical cancer was the third, with more new cases in 2020 in women. Lung cancer was the fourth with more new cases in 2020 in men and women, and esophageal cancer was the sixth with more new cases in men. With regard to mortality, data from 2018 indicate that the cancers selected for this study are among the top five in number of deaths. These values refer to both genders for lung cancer, to female strata for breast and cervical cancers, and to the male stratum for esophageal cancer [11]. Although brain cancer does not have a high frequency in Brazil and metastatic brain tumors are more frequent than primary brain tumors, as several studies have evaluated their relationship with exposure to electromagnetic fields, we have included the results of the analysis of this type of cancer in Appendix A.

Census population data [12] and GDP were also collected from the DATASUS website [10]. The number of RBSs and the year they were implemented in each capital were collected from Telecommunication Service System [2].

People’s exposure times were calculated according to birth and death years. The annual RBS radiofrequency exposure was calculated by summing the number of RBS implementations in each year multiplied by the people’s exposure time. The total exposure was calculated from the sum of annual exposures.

A map with charts was built using the mortality rate per square kilometer (km²) and the median of RBS radiofrequency exposure in the 2010–2017 period.

## 3. Statistical Analysis

We calculated the median and interquartile range of mortality rate per 100,000 according to the levels of explanatory variables. The Kruskal–Wallis test was used to access the statistical differences between groups.

Multilevel Poisson regression models were used to estimate the risk-adjusted mortality. The response variable was death by cancer, and the fixed effects were the logarithm transformation of RBS radiofrequency exposure, gender, age group, and death year. We also included an offset with the logarithm of population size. The random effects included capital city (intercept), square root transformation of GDP (slope), and capital’s area per km² (slope). When the response variables were death by breast and cervix cancer, the gender was not included as a fixed effect, as only females were investigated.

The abovementioned logarithmic transformations and the square root transformation were used to normalize the distribution of variables. We used R-Project version 3.6.1 (R Foundation, Vienna, Austria) and ArcGis version 10.5 (Environmental Systems Research Institute, Redlands, CA, USA) to perform the analysis.

## 4. Results

For all cancers and for the specific types investigated (breast, cervix, lung, and esophagus cancers), the higher the exposure to RBS radiofrequency, the higher the median of mortality rate. In capitals where RBS radiofrequency exposure was higher than 2000/antennas-year, the median of the breast cancer mortality rate was 27.33/100,000, while for all cancers, it was 111.68/100,000 (Table 1).

Females showed the highest median of mortality rate for all cancers but specifically for lung and esophagus cancers; the highest median of mortality rate was observed in males (4.31/100,000 and 0.45/100,000, respectively) (Table 1).

For all cancers and for the specific types investigated, the higher the age group, the higher the median of mortality rate. Lung and breast cancers showed high medians of mortality rate (159.40/100,000 and 91.18/100,000, respectively) (Table 1).

The median of mortality rate for all types of cancer decreased from 68.76/100,000 to 61.87/100,000 over the period. For breast, cervix, lung, and esophagus cancers, it showed slight variations over the period, around 17/100,000, 7/100,000, 4/100,000, and lower than one per 100,000, respectively (Table 1).

In the adjusted analysis, the results showed that the higher the logarithm of RBS radiofrequency exposure, the higher the cancer mortality rate. The highest adjusted risk was observed for cervix cancer (Rate Ratio (RR) = 2.18) (Table 2).

A multilevel Poisson model was used to estimate the risk of cancer mortality. Except for breast and cervix cancers, which were estimated only for women, every adjusted models included as fixed effects the variables sex, logarithm of RBS, age group, and death year. The variables included as random effects were capital (intercept), GDP (slope), and area/Km^2^ (slope). The offset of the population was also included in the models.

Males showed the highest adjusted risk of lung, esophagus, and all types of cancer (Table 2), although the median of mortality rate for all cancers was higher for females in the bivariate analysis (the results are shown in Table 1).

As expected, there was an increasing trend in the adjusted risk of cancer mortality in the older the age group. Compared to people younger than 30 years old, the adjusted risks were 297.55, 53.88, 1250.63, 2154.44, and 164.61 for breast, cervix, lung, esophagus cancer, and all cancers, respectively (Table 2).

For cervix cancer and all types of cancers, there was a decreasing trend in the adjusted risk of cancer mortality for more recent periods. For lung and esophagus cancers, this trend is observed from 2014–2015 period (Table 2).

The inclusion of random effects was significant in every models for the following effects: “capital” (intercept) and “square root of GDP” (slope). However, for the “area of the capital” (slope), it was significant only for esophagus cancer. For all models, the greatest standard deviation of random effects was observed for the “capital” (intercept) (Table 2).

The spatial descriptive analysis showed that the highest median of RBS radiofrequency exposure was observed in Florianópolis (South of Brazil) (44.23 antennas-year/km²). Florianópolis also has the highest mortality rate per km² for all types of cancer and specifically for lung and breast cancer (0.09/100,000, 0.31/100,000, and 0.93/100,000, respectively). Recife (Northeast) and Belo Horizonte (Southeast) showed medians of RBS radiofrequency exposure higher than 20 antennas-year/km², and their mortality rates per km² for all cancers were 0.29/100,000 and 0.19/100,000, respectively. Vitoria (Southeast), Teresina, Fortaleza, Natal, and Aracaju (both in Northeast) showed medians of RBS radiofrequency exposure higher than 10 antennas-year/km², and mortality rate per km² for all types of cancer were 0.60/100,000, 0.49/100,000, 0.21/100,000, 0.35/100,000, and 0.38/100,000, respectively (Figure 1).

## 5. Discussion

The biological effects of exposure to electromagnetic fields were investigated in some studies, mainly experimental studies with mice. The authors point out the following effects: an increase in the calcium efflux in human neuroblastoma cells, impairing cellular functions [13]; changes in the immune system [14]; a decrease in reproductive function [15]; an increase in serum testosterone levels [16]; an increase in permeability of the blood–brain barrier, which protects the brain from toxic substances, bacteria, and viruses [17]; and damage to cell DNA [18,19,20].

The evidence of radiofrequency radiation carcinogenesis has increased since 2011. Some animal studies suggest that exposure to electromagnetic fields accelerate the development of sarcoma colonies in the lung, mammary tumors, skin tumors, hepatomas, and sarcomas [21,22]. This study detected an increase in the rate of death by cancer in capitals where there is a greater exposure to electromagnetic fields emanating from radio base stations. Studies made in Stockholm (Sweden) indicate that high levels of environmental radiofrequency radiation are quite present in residential and commercial areas [23,24,25]. In the United Kingdom, at the beginning of 2009, there were 51,300 RBSs and two thirds were installed in existing buildings or other structures [9].

Dode et al., 2011, pointed that electromagnetic fields from telecommunication systems is an important environmental problem nowadays [8]. The authors detected 6724 deaths by neoplasia occurring within an area of 500 m from the RBS and 320 deaths within an area between 500 and 1000 m. The mortality rate within an area of 500 m was 34.76 per 10,000 inhabitants, while within an area of 1.000 m, the rate was 32.78 [8].

Although in the present study, we investigate all cancers, we also investigated breast, cervix, lung, and esophagus cancers separately. In a mortality study performed in Brazil, breast and lung cancers were among the main cancers related to radiofrequency electromagnetic fields from RBS radiofrequency exposure [8].

Breast and cervix cancer have cure rates of around 95% when diagnosed early [26]. Mortality from breast cancer continues to increase in Brazilian capitals, while mortality from cervical cancer remains stable, unlike what occurs in developed countries, in which mortality for these cancers is decreasing. Lung cancer has less chances to be cured when detected in the early stages (56%) [26]. Esophageal cancer is difficult to detect early. In most cases, the signs and symptoms only appear in more advanced stages of the disease [26].

Despite the advance in knowledge about cancer, not all countries seem to benefit from this advance. This is the case of low- and middle-income countries, where a significant portion of the population does not have access to diagnosis and treatment, decreasing their chances of survival.

In the present study, we detected that the higher the exposure to radiofrequency electromagnetic fields from RBSs, the higher cancer mortality is. A study conducted in Israel also found that low-frequency electromagnetic fields contribute to breast cancer development [27,28,29]. Others studies also referred to the relationship between cancer and radiofrequency electromagnetic fields [30,31], including in animal studies [32].

In addition to exposures to radiofrequency electromagnetic fields, we have to consider other factors that contribute to the increase in cancer incidence and mortality. In Brazil, about 70% of the population depends on public health [33], and there are difficulties accessing cancer diagnosis and treatment in public health services. Opportunistic screening is still adopted, performed only when the patient in the risk group comes to the health service and there is difficulty starting cancer treatment within 60 days, as required by Brazilian law [34]. The consequences of these problems are the worsening of the disease and the high number of preventable deaths.

The main risk factor for lung cancer is tobacco consumption, which is higher for males [35]. Tobacco consumption has been decreasing gradually in Brazil from 1980 to 2013 [36], and this decline may have contributed in some way to reducing lung cancer mortality in men over time [37]. The main risk factors for esophageal cancer are the high intake of hot drinks [38], alcoholic beverages, and tobacco; low ingestion of fruit and vegetable; and exposure to occupational agents like benzene, silica, and asbestos [39]. Family history is one of the most important breast cancer risk factors [40]. However, there are many other risk factors, such as aging, genetic mutations, reproductive history, dense breasts, past history of breast disease, previous treatment with radiotherapy, sedentary lifestyle, overweight or obesity after menopause, alcohol intake, and use of hormones and some oral contraceptives [41]. Cervix cancer risk factors are associated with the risk of Human Papillomavirus (HPV) infection. A high number of pregnancies and no regular preventive colpocytology are pointed out as risk factors to cervix cancer [42].

In the present study, a capital located in the south showed the highest RBS radiofrequency exposure and the highest mortality by cancer (Florianópolis). In fact, other studies have also reported high rates of cancer in this capital [43,44,45,46,47].

Our results showed that, in general, men had a higher mortality rate of esophageal and lung cancer and that this rate increases with age. In fact, the scientific literature corroborates these results [48,49,50,51,52].

In order to keep the cellular sets running, the radiofrequency transmitters installed on the tops, roofs, and façades of buildings and residences emit electrical and magnetic fields 24 h a day. However, it is known by scientific knowledge that the nonthermal magnetic component can penetrate deeper into the body than the electrical one [53].

A person can stand at a fixed distance to an RBS and be exposed to 100% of the maximum permissible exposure or 5% of it depending on the antenna height and the bystander altitude. Therefore, people living in the upper floors of a building located in front of the antennas receive radiofrequency corresponding to 100% of the maximum permissible exposure [54]. Those data were confirmed in the Post-Graduation Project conducted at the Federal University of Minas Gerais (UFMG), Brazil, based on measures made in the capital of the state inside 400 residences located near the RBS from 2015 to 2019, measurements made by “MRE Engenharia—Medição de Radiações Eletromagnéticas” [55].

The measured values of the electrical and magnetic fields have shown more human exposure to electromagnetic radiation in an area within a 500 m radius from the transmitter antennas of cellular telephony [8]. To avoid hazards to human health, the safest solution would be to switch off the RBS in an area within a 500 m radius from residences, workplaces, hospital areas, kindergartens, and buildings.

As a limitation, it is important to note that this study used cancer data from national Brazilian sources, which can provide underestimated rates at different levels according to the region. For example, a study conducted in northern Brazil found a large proportion of deaths classified as unspecified uterine cancer. After the proportional redistribution of these deaths, there was an increase of 46% in the average cervical cancer mortality rates [56]. Another study conducted in a northeastern region of Brazil highlights that, within the older age group, the number of deaths before and after correction showed a significant variation, especially for breast cancer, where variation reached 54% [57].

Still as limitations of the study, we highlight two more points. As this is an ecological study and due to the unavailability of individual dwelling time data, the time that each individual lived close to an RBS could not be accessed. The possible migratory movements could also not be considered for calculation of the amount of exposure to RBS radio frequency throughout life in the resident population. This was calculated only according to birth, death year, and the year in which the RBS was implemented. However, people could have been born in another city and then migrated to the capital where the death was recorded. The other point is the proximity of stations to places of residence that interfere with the level of exposure of individuals. As it is an ecological study in which the unit of analysis is the capital, this study did not take into account the distances between stations and homes.

## 6. Conclusions

The balance of our results indicates that the exposure to radiofrequency electromagnetic fields from an RBS increases the rate of mortality by all cancers and specifically by breast, cervix, lung, and esophageal cancers. These conclusions are based on the fact that the findings of this study indicate that, the higher the RBS radiofrequency exposure, the higher the cancer mortality rate, especially for cervix cancer (adjust RR = 2.18). The spatial analysis showed that the highest RBS radiofrequency exposure was observed in a city located in the southern region of Brazil, which also showed the highest mortality rate for all types of cancer and specifically for lung and breast cancers.

Environmental pollution caused by nonionizing electromagnetic fields increases continuously. The location of RBSs is still a controversial field with regard to their regulation. There are numerous RBSs installed in residential areas, including on their roofs. Some epidemiological studies indicate an increased risk of cancer close to RBSs. It is important that further epidemiological investigations are conducted to elucidate the role of the environment in radiofrequency signals on carcinogenesis processes.

## Figures and Tables

**Figure 1 ijerph-18-01229-f001:**
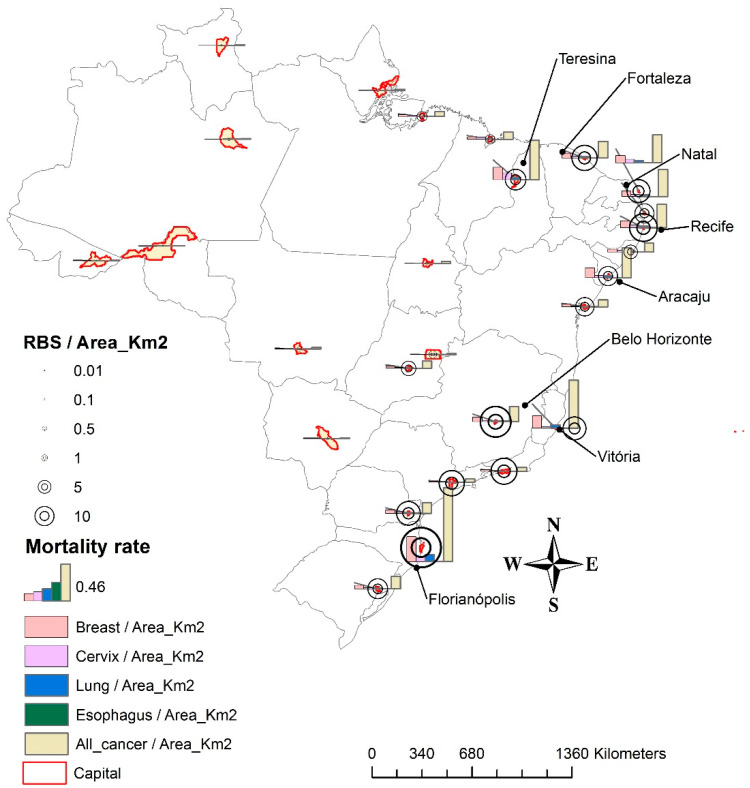
Distribution of the cancer mortality rate in each capital and their experience of exposure to radio base stations radiofrequency, 2010–2017. RBS = median of the number of radio base station radiofrequency exposure (RBS radiofrequency exposure = sum of the number of RBS in each year multiplied by the exposure time). RBS/Area_Km2 = median of the number of RBS per Km^2^. We used the median of mortality rate. Breast/Area_K2, Cervix/Area_K2, Lung/Area_K2, or Esophagus/Area_K2 = median mortality rate for a specific cancer type per Km^2^.

**Table 1 ijerph-18-01229-t001:** Descriptive analysis of cancer mortality in Brazilian capitals.

	Breast	Cervix	Lung	Esophagus	All Cancers
	Median/10^5^ (IQR)	Median/10^5^ (IQR)	Median/10^5^ (IQR)	Median/10^5^ (IQR)	Median/10^5^ (IQR)
RBS-sign	*	*	*	*	*
≤500	0.00 (0.00)	0.00 (0.00)	0.00 (0.00)	0.00 (0.00)	7.30 (44.94)
501–1000	1.16 (27.11)	2.74 (26.30)	0.00 (38.97)	0.00 (0.00)	26.32 (382.14)
1001–2000	20.12 (54.53)	7.38 (25.79)	4.47 (63.42)	0.00 (8.74)	71.95 (500.43)
>2000	27.33 (63.06)	9.56 (16.43)	9.58 (76.46)	1.62 (14.21)	111.68 (552.78)
Sex-sign			*	*	*
Female			3.77 (46.88)	0.00 (3.17)	75.31 (360.87)
Male			4.31 (98.82)	0.45 (22.06)	56.49 (540.97)
Age group-sign	*	*	*	*	*
<30	0.00 (0.00)	0.00 (0.00)	0.00 (0.00)	0.00 (0.00)	5.75 (4.53)
30–49	9.89 (13.56)	6.75 (7.31)	1.81 (4.39)	0.00 (1.13)	38.59 (44.90)
50–69	43.43 (20.19)	15.02 (14.71)	34.08 (42.50)	6.75 (16.28)	258.79 (240.76)
≥60	91.18 (64.51)	27.35 (37.02)	159.40 (159.63)	20.31 (39.68)	1178.11 (1012.72)
Year-sign	NS	NS	NS	NS	NS
2010–2011	16.95 (52.66)	6.29 (19.36)	4.44 (64.91)	0.00 (8.87)	68.76 (508.70)
2012–2013	15.98 (56.94)	6.42 (19.09)	4.13 (66.30)	0.00 (10.29)	65.09 (501.19)
2014–2015	17.36 (56.05)	8.29 (19.52)	4.13 (65.15)	0.00 (9.54)	65.56 (491.10)
2016–2017	18.01 (52.08)	7.62 (16.66)	3.54 (65.52)	0.00 (8.22)	61.87 (444.41)

RBS = exposure to radio base stations (antennas-year); IQR = interquartile range; sign = statistical significance − significance. * *p*-value < 0.001 and NS, *p*-value > 0.05.

**Table 2 ijerph-18-01229-t002:** Adjusted risk of cancer mortality in Brazilian capitals.

	Breast		Cervix		Lung		Esophagus		All Cancers	
	RR	Sign	RR	Sign	RR	Sign	RR	Sign	RR	Sign
Fixed effects										
Log RBS	1.25	***	2.18	***	1.14	***	1.18	**	1.15	***
Sex										
Female					1.00		1.00		1.00	
Male					1.97	***	4.88	***	1.42	***
Age group										
<30	1.00		1.00		1.00		1.00		1.00	
30–49	37.59	***	13.82	***	20.11	***	73.84	***	6.06	***
50–69	132.29	***	30.74	***	323.80	***	876.50	***	40.73	***
≥60	297.55	***	53.88	***	1250.63	***	2154.44	***	164.61	***
Year										
2010–2011	1.00		1.00		1.00		0.00		1.00	
2012–2013	0.97	NS	0.78	***	0.97	NS	0.96	NS	0.98	*
2014–2015	0.96	NS	0.62	***	0.93	**	0.88	***	0.95	***
2016–2017	0.81	**	0.46	***	0.84	***	0.76	***	0.84	***
Random effects										
	Std Dev				Std Dev		Std Dev		Std Dev	
Capital (intercept)	0.61	***	1.55	***	0.19	***	0.86	***	0.28	***
Sqrt GDP (slope)	0.00	***	0.01	***	0.00	***	0.00	***	0.00	***
Area/Km^2^ (slope)	0.00	NS	0.00	NS	0.00	NS	0.00	*	0.00	NS
Deviance	12274		8345		24732		10364		100918	

Sqrt GDP = square root transformation of gross domestic product per capita; RR = rate ratio; Std Dev = standard deviation; Log RBS = logarithm transformation of radio base station radiofrequency exposure (RBS radiofrequency exposure = sum of the number of RBS in each year multiplied by the exposure time); sign = statistical significance − significance. *** *p*-value < 0.001; ** *p*-value < 0.01; * *p*-value < 0.05; and NS, *p*-value > 0.05.

## Data Availability

The data presented in this study are available on request from the corresponding author.

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
