# Peer review of "The Effect of Continuous Low-Intensity Exposure to Electromagnetic Fields from Radio Base Stations to Cancer Mortality in Brazil"

_ijerph, 2021, doi:10.3390/ijerph18031229_

Round 1

Reviewer 1 Report

I have read the manuscript entitled "The Effect of Continuous Low Intensity Exposure to Electromagnetic Fields from Radio Base Stations to Cancer Mortality in Brazil" authored by Nádia Cristina Pinheiro Rodrigues and co-workers. The aim of this study was to estimate the rate of death by cancer, according to Radio Base Stations radiofrequency exposure, especially for the types of breast, cervix, lung and esophagus cancer.

In my opinion the electromagnetic field is not the only environmental factor that affects human health. Carrying out the analysis, there should be paid an attention to other factors that may affect the incidence of cancer. When analyzing lung cancer incidence, it is extremely important to state whether the patients were long-term smokers. For example, esophageal cancer can be caused by high-alcohol consumption, smoking, acid reflux, or a poor diet.  In breast and cervical cancer, it is important whether the patient has previously undergone hormonal therapy, oral contraceptives and pregnancy. In all cases it should be taken into account whether the cancer was the only cause of death, whether there were comorbidities.

In my opinion the date of birth and death does not authorize the determination of the time of exposure. The only factor to determine the exposure time is timing residence in a location close to the RBS.

In addition, the publication of the paper in a journal of high international importance requires discussion not only with dry statistics, but also at least a few studies in which the authors present the results of the impact of EMF RF on animal cells and tissues, for example measured by in vitro method. In my opinion, after completing the comments made, the paper will gain a lot in value and then I will recommend it for approval for publication.

Author Response

Reviewer 1

I have read the manuscript entitled "The Effect of Continuous Low Intensity Exposure to Electromagnetic Fields from Radio Base Stations to Cancer Mortality in Brazil" authored by Nádia Cristina Pinheiro Rodrigues and co-workers. The aim of this study was to estimate the rate of death by cancer, according to Radio Base Stations radiofrequency exposure, especially for the types of breast, cervix, lung and esophagus cancer.

In my opinion the electromagnetic field is not the only environmental factor that affects human health. Carrying out the analysis, there should be paid an attention to other factors that may affect the incidence of cancer. When analyzing lung cancer incidence, it is extremely important to state whether the patients were long-term smokers. For example, esophageal cancer can be caused by high-alcohol consumption, smoking, acid reflux, or a poor diet.  In breast and cervical cancer, it is important whether the patient has previously undergone hormonal therapy, oral contraceptives and pregnancy. In all cases it should be taken into account whether the cancer was the only cause of death, whether there were comorbidities.

Answer: We added in the discussion the other risk factors involved in each type of cancer.

Line 289

“In addition to exposures to radiofrequency electromagnetic fields, we have to consider the other factors that contribute to the increase in cancer incidence and mortality. In Brazil, about 70% of the population depends on public health (33), and there are difficulties in accessing cancer diagnosis and treatment in public health services. It is still adopted the opportunistic screening, performed only when the patient in the risk group comes to the health service and there is difficulty of starting cancer treatment in 60 days, as required by Brazilian law (34). The consequences of these problems are the worsening of the disease and the high numbers of preventable deaths.

The main risk factor for lung cancer is tobacco consumption, which is higher for males (35). Tobacco consumption has been decreasing gradually in Brazil from 1980 to 2013 (36), and this decline may have contributed in some way to reducing lung cancer mortality in men over time (37). The main risk factors for esophageal cancer are the high intake of hot drinks (38), alcoholic beverages and tobacco, low ingestion of fruit and vegetable and exposure to occupational agents like benzene, silica and asbestos (39). Family history  is one of the most important breast cancer risk factors (40). However, there are many other risk factors, such as aging, genetic mutations, reproductive history, dense breasts, past history of breast disease, previous treatment with radiotherapy, sedentary lifestyle, overweight or obesity after menopause, alcohol intake, use of hormones and some oral contraceptives (41). Cervix cancer risk factors are associated with the risk of Human Papillomavirus (HPV) infection. High number of pregnancies and no regular preventive colpocytology are pointed out as risk factors to cervix cancer (42).”

In my opinion the date of birth and death does not authorize the determination of the time of exposure. The only factor to determine the exposure time is timing residence in a location close to the RBS.

Answer: Yes, in fact the ideal would be to know how long each individual has lived close to RBS. As we do not have this information, we point this issue out in the limitations in the last paragraph of the discussion.

Line 348

“Still as limitations of the study, we highlight two more points. As this is an ecological study and also due to the unavailability of individual dwelling time data, the time that each individual lives close to RBS could not be accessed. The possible migratory movements could also not be considered for the calculation of the amount of exposure to RBS radio frequency throughout life in the resident population. This was calculated only according to birth, death year and the year, in which RBS was implemented. However, people could have been born in another city and then migrated to the capital where the death was recorded. The other point is the proximity of stations to places of residence that interfere with the level of exposure of individuals. As it is an ecological study, whose unit of analysis is the capital, this study did not take into account the distances between stations and homes.”

In addition, the publication of the paper in a journal of high international importance requires discussion not only with dry statistics, but also at least a few studies in which the authors present the results of the impact of EMF RF on animal cells and tissues, for example measured by in vitro method. In my opinion, after completing the comments made, the paper will gain a lot in value and then I will recommend it for approval for publication.

Answer: We have included some more studies.

Line 243

“The biological effects of exposure to electromagnetic fields have been investigated in some studies, mainly experimental studies with mice. The authors point out the following effects: increase of calcium efflux in human neuroblastoma cells, impairing cellular functions (13); changes in the immune system (14); decrease of reproductive function (15); increase in serum testosterone levels (16); increase of permeability of the blood-brain barrier, which protects the brain from toxic substances, bacteria and viruses (17); and damage to cell DNA (18-20).

The evidence on radiofrequency radiation carcinogenesis has increased, since 2011. Some animal studies suggest that exposure to electromagnetic fields accelerate the development of sarcoma colonies in the lung, mammary tumors, skin tumors, hepatomas, and sarcomas (21, 22).”

Reviewer 2 Report

I would like to see more information about why the authors chose cervical cancer as a cancer to be studied.  I don't know whether it is possible to know the HPV status of the people with cervical cancer.   Does the proximity to a radiation source suggest less ability for women to have screening mammograms - hence contributing to the breast cancer rates, as well as the use of HPV vaccination and / or knowledge of HPV status.

I would also like to know, even if this was in a table or appendix if the authors found any increase in brain cancers

The scientific methods seem sound and I would recommend the article for publication.

Author Response

Reviewer 2

I would like to see more information about why the authors chose cervical cancer as a cancer to be studied.  I don't know whether it is possible to know the HPV status of the people with cervical cancer.   Does the proximity to a radiation source suggest less ability for women to have screening mammograms - hence contributing to the breast cancer rates, as well as the use of HPV vaccination and / or knowledge of HPV status.

Answer: We have included cervical cancer because it is among the most frequent in Brazil, however, we added to the discussion the other risk factors involved in each type of cancer, especially those related to cervical cancer.

Line 289

“In addition to exposures to radiofrequency electromagnetic fields, we have to consider the other factors that contribute to the increase in cancer incidence and mortality. In Brazil, about 70% of the population depends on public health (33), and there are difficulties in accessing cancer diagnosis and treatment in public health services. It is still adopted the opportunistic screening, performed only when the patient in the risk group comes to the health service and there is difficulty of starting cancer treatment in 60 days, as required by Brazilian law (34). The consequences of these problems are the worsening of the disease and the high numbers of preventable deaths.

The main risk factor for lung cancer is tobacco consumption, which is higher for males (35). Tobacco consumption has been decreasing gradually in Brazil from 1980 to 2013 (36), and this decline may have contributed in some way to reducing lung cancer mortality in men over time (37). The main risk factors for esophageal cancer are the high intake of hot drinks (38), alcoholic beverages and tobacco, low ingestion of fruit and vegetable and exposure to occupational agents like benzene, silica and asbestos (39). Family history  is one of the most important breast cancer risk factors (40). However, there are many other risk factors, such as aging, genetic mutations, reproductive history, dense breasts, past history of breast disease, previous treatment with radiotherapy, sedentary lifestyle, overweight or obesity after menopause, alcohol intake, use of hormones and some oral contraceptives (41). Cervix cancer risk factors are associated with the risk of Human Papillomavirus (HPV) infection. High number of pregnancies and no regular preventive colpocytology are pointed out as risk factors to cervix cancer (42).”

I would also like to know, even if this was in a table or appendix if the authors found any increase in brain cancers

Answer: The results of the analysis of data related to brain cancer were added in the appendix.

Line 136

“…Although brain cancer does not have a high frequency in Brazil and metastatic brain tumors are more frequent than primary brain tumors, as several studies have evaluated their relationship with exposure to electromagnetic fields, we have included the results of the analysis of this type of cancer in Appendix A.”

The scientific methods seem sound and I would recommend the article for publication.

Reviewer 3 Report

The authors of this manuscript have done an excellent study on the effect of radiations exposure RBS on cancer mortality in a particular geographical location. It would be interesting to see how such analysis varies in different locations. The study would have been more significant if it would include how this exposure affects cancer initiation and progression as well and not just mortality rate. Here are few other comments to be addressed:

  1. In line 17 in the abstract, instead of 'according to', isn't it 'as a result of RBS radiofrequency exposure?
  2. Line 22, instead of writing 'Results.' it can be rephrased into a sentence begining as--Our results/findings show that..

  3.  In the background section, it would be appropriate to elaborate why the study aimed to specifically look at certain cancer types and not the others.
  4. In line 82, it is mentioned that the study included radiofrequency exposure in 2010-2017 time period. Is there an explanation on why this time frame was chosen and not till a more recent time frame till 2019 or 2020?

  5. Fig 1 representation could be improved for clarity

Author Response

Reviewer 3

The authors of this manuscript have done an excellent study on the effect of radiations exposure RBS on cancer mortality in a particular geographical location. It would be interesting to see how such analysis varies in different locations. The study would have been more significant if it would include how this exposure affects cancer initiation and progression as well and not just mortality rate. Here are few other comments to be addressed:

Answer: We try to show in Figure 1 how mortality and exposure varies in each capital. I agree that it would be interesting to also show incidence data, but these data are not available in all Brazilian capitals.

In line 17 in the abstract, instead of 'according to', isn't it 'as a result of RBS radiofrequency exposure?

  1. Line 22, instead of writing 'Results.' it can be rephrased into a sentence begining as--Our results/findings show that..

Answer: "Results", refers to the topic title. If I choose to remove this topic, we will also have to remove the remaining topics from the abstract - Background, Methods, Conclusion. In this way we chose to keep the topics.

  1. In the background section, it would be appropriate to elaborate why the study aimed to specifically look at certain cancer types and not the others.

Answer: We add explanation at the end of the introduction and in the methods.

Background

Line 105

“Despite the scarce knowledge on this subject, there are few resources allocated to investigating the role of exposure to electromagnetic fields from RBS on human health. In the United States, for example, until 2010, no funding had been reserved by government agencies to study the possible health effects on people living near RBS 9. This study aims to estimate the rate of death for cancer, according to RBS radiofrequency exposure, especially by breast, cervix, lung and esophagus cancer, especially for breast, cervix, lung and esophagus cancer, which are among the most common cancers in Brazil in men, women or both sexes.”

Methods

Line 125

“…The choice of these specific types of cancer for this study took into account their high frequency of new cases in women or in men. Current statistics from 2020 from the National Cancer Institute show that breast cancer had the highest number of new cases in 2020 for women (about 66,000 cases, corresponding to about 30% of cases). Cervical cancer was the third with more new cases in 2020 in women. Lung cancer was among the four with more new cases in 2020 in men and women and esophageal cancer was the sixth with more new cases in men. Regard to mortality, data from 2018 indicate that the cancers selected for this study are among the top five in number of deaths. These values refer to both genders for lung cancer, to female stratum for breast and cervical cancers and to male stratum for esophageal cancer….”

  1. In line 82, it is mentioned that the study included radiofrequency exposure in 2010-2017 time period. Is there an explanation on why this time frame was chosen and not till a more recent time frame till 2019 or 2020?

Answer: The period was stipulated until 2017 because this was the last year for which data were available.

  1. Fig 1 representation could be improved for clarity

Answer: The figure's footer has been organized and other explanations have been added to this footer.

Line 232

“Figure 1. Distribution of cancer mortality rate in each capital and their experience of exposure to Radio Base Stations radiofrequency, 2010-2017.

RBS = Median of Radio Base Station radiofrequency exposure (RBS radiofrequency exposure = sum of the number of RBS in each year multiplied by the exposure time).

RBS/Area_Km2 = mediana do número de RBS por Km2

We used the median of mortality rate.

Breast/Area_K2 or Cervix/Area_K2 or Lung/Area_K2 or Esophagus/Area_K2   = median mortality rate for a specific cancer type per Km2

Round 2

Reviewer 1 Report

The revised article takes into account all comments indicated by the reviewer. In my opinion, it can be accepted for printing in this form. The presented results can be used by government administration units in assessing public health.